# Five Unreported Ketone Compounds—Penicrustones A–E—From the Endophytic Fungus *Penicillium crustosum*

**DOI:** 10.3390/microorganisms12112195

**Published:** 2024-10-30

**Authors:** Dongmei Lin, Lian Yang, Jin Yang, Feixing Li, Xiuming Cui, Xiaoyan Yang

**Affiliations:** 1Faculty of Life Science and Technology, Kunming University of Science and Technology, Kunming 650500, China; 20222118071@stu.kust.edu.cn (D.L.); 20212118109@stu.kust.edu.cn (L.Y.); 20232118083@stu.kust.edu.cn (J.Y.); 20222218039@stu.kust.edu.cn (F.L.); 20120094@kust.edu.cn (X.C.); 2Yunnan Key Laboratory of Sustainable Utilization of Panax Notoginseng, Kunming 650500, China

**Keywords:** *Penicillium crustosum*, fungus, penicrustone, activity, X-Ray diffraction

## Abstract

Five unreported ketone compounds—penicrustones A–E—were isolated from the solid fermentation of the endophytic fungus *Penicillium crustosum*. Their structures were elucidated on the basis of extensive spectroscopic analysis. Their absolute configurations were determined via ECD calculations and single-crystal X-Ray crystallography. All compounds were evaluated for their antimicrobial and antitumor activities. Compounds **4** and **5** showed moderate inhibitory effects on *Micrococcus luteus*, with MIC values of 12.5 and 25.0 μg/mL, respectively. In addition to this, compound 4 also showed cytotoxicity on tumor cell lines KTC-1 and Hela, with IC_50_ values of 4.28 and 4.64 μg/mL, respectively.

## 1. Introduction

Endophytic fungi are widely present in various tissues and organs of healthy plants. Due to their unique living environment and survival strategy, they have not only acquired the ability to symbiotically live with plants, but also possess strong biosynthetic capabilities, which allow them to produce diverse secondary metabolites. As a result, endophytic fungi have become an important resource of natural products. Among them, the endophytic fungi of the *Penicillium* genus play a particularly important role in providing natural active compounds. In recent years, many structurally diverse secondary metabolites have been isolated from the genus *Penicillium*, including phenylenone derivatives [1], benzannulated 6,6-spiroketal [2], citrinin derivatives [3], alkaloids [4,5,6], terpenoids [7,8], and polyketones [9]. Some of these compounds showed IDO-1 inhibitory and mycobacterium tuberculosis protein tyrosine phosphatase B inhibitory properties, as well as antibacterial activities and more [1,2,6]. *Penicillium* is one of the largest fungal genera currently found in living organisms [10]; it has emerged as a well-known source of antibiotics. However, through metabolomics analysis, there are still numerous other medicinal potentials that await development [11].

*Penicillium crustosum* is a fungus that is widely found in various natural environments, including soil, plants, oceans, and so on. It has the ability to produce a diverse range of bioactive secondary metabolites. While some studies have documented the isolation of alkaloids [12,13], aromatic polyketones [14,15], terpenoids [16], and other compounds from the fermentation of this fungus, there is still significant potential for the exploration of new secondary metabolites from this species. We investigate the secondary metabolites of the fungus *P. crustosum*, and five new phenylethyl ketone compounds—penicrustones A–E—were isolated (Figure 1). The present paper describes the isolation and structure elucidation of the penicrustones A–E; their antimicrobial activity was tested against seven human pathogens and they were cytotoxically evaluated against KTC-1 and Hela cell lines.

## 2. Materials and Methods

### 2.1. Fungal Material

The fungus *P. crustosum* was provided by the Faculty of Life Science and Technology, Kunming University of Science and Technology. A voucher specimen of this strain was deposited at the Yunnan Key Laboratory of *Panax notoginseng*, with the strain certificate number YXY20210413Q4. The inoculation of *P. crustosum* in PDA culture involved a three-day cultivation period at 25 °C for rejuvenation. Once the mycelium was fully developed in the culture medium, it was transferred to PDB medium and incubated on a shaking table at 25 °C and 150 rpm for three days, serving as “seeds” for further fermentation in rice medium. The “seeds” were then inoculated into 500 mL glass tissue culture bottles, each containing 150 g of rice and 150 mL of a 2% glucose solution. A total of 600 bottles were cultured at 25 °C in the dark for 60 days.

### 2.2. General Experimental Procedures

Melting points were determined using the X-4 micromelting point apparatus (Shanghai INESA Physico-Optical Instrument Co., Ltd., Shanghai, China). Optical rotations were measured with a Rudolph Autopol VI (Rudolph, Munich, Germany). Electronic circular dichroism (ECD) spectra were obtained on a Chirascan V100 (Applied Photophysics, Surrey, UK). UV spectra were recorded using a Shimadzu UV-2401PC spectrophotometer (Shimadzu, Kyoto, Japan). Infrared (IR) spectra were measured with a Bruker PMA50 spectrometer (Bruker BioSpin Group, Karlsruhe, Germany). Single-crystal X-Ray diffraction analysis was conducted using a Bruker D8 QUEST (Bruker AXS GmbH, Karlsruhe, Germany). Nuclear magnetic resonance (NMR) spectra were acquired on a Bruker AV 600 spectrometer (Bruker Corporation, Karlsruhe, Germany), with compounds calibrated using tetramethylsilane (TMS). Electrospray ionization mass spectrometry (ESI-MS) and high-resolution ESI-MS (HR-ESI-MS) were performed using a UPLC-IT-TOF spectrometer (Shimadzu Corporation, Kyoto, Japan). Semi-preparative high-performance liquid chromatography (HPLC) was conducted on a Shimadzu LC-6AD equipped with an ODS-A column (5 μm, 10.0 mm × 250 mm, YMC Co., Ltd., Kyoto, Japan). Chiral HPLC analysis and separation were carried out using a Daicel Chiralcel OJ-H column on an Agilent 1260 system (5 μm, 4.6 mm × 250 mm, Daicel Chiral Technologies Co., Ltd., Shanghai, China). Column chromatography (CC) was performed with silica gel (Qingdao Haiyang Chemical Co., Ltd., Qingdao, China) and Sephadex LH-20 (Amersham Biosciences, Uppsala, Sweden). Thin chromatography (TLC) analysis of various fractions was conducted using silica gel GF 254 (Qingdao Marine Chemical Factory, Qingdao, China).

### 2.3. Extraction and Isolation

The culture of *P. crustosum* was extracted three times with CH_2_Cl_2_/MeOH (1:1, *v*/*v*) at room temperature to give an crude extract, before being extracted three times with EtOAc to afford the extract (705.0 g). The extract was subjected to CC on silica gel (100–200 mesh) and eluted with a petroleum ether (PE)/EtOAc (*v*/*v*, 1:0 to 0:1) gradient to obtain six fractions A–F. Fr.C (135.7 g) was separated using MPLC over a C-18 column eluted with MeOH/H_2_O (80–100%) to give five subfractions (Fr.C1–5). Fr.C4 (0.11 g) was isolated via CC over a silica gel eluted with (PE)/EtOAc (*v*/*v*, 10:1 to 1:1) to give compound **4** (34.0 mg). Fr.D (107.5 g) was separated using MPLC over a C-18 column eluted with MeOH/H_2_O (70–100%) to give four subfractions (Fr.D1–4). Fr.D2 (0.42 g) was subjected to CC using MeOH on Sephadex LH-20 to obtain three subfractions (Fr.D2a–c). Fr.D2b (0.11 g) was subjected to silica gel CC eluting with (PE)/EtOAc (*v*/*v*, 10:1) to give compound **1** (2.0 mg). Using CH_2_Cl_2_/MeOH (1:1, *v*/*v*) on Sephadex LH-20 to treat Fr.D3 (0.37 g), it was purified to give compound **2** (4.8 mg). The separation of Fr.E (181.3 g) using MPLC with a MeOH/H_2_O (40–100%) step gradient obtained four sub components (Fr.E1–6). Fr.E2 (1.8 g) was eluted with MeOH via Sephadex LH-20 CC to obtain three subfractions (Fr.E2a–d). Fr.E2b (0.53 g) was subjected to alkaline Al_2_O_3_ CC eluting with CH_2_Cl_2_ to obtain compound **3** (25.0 mg). Fr.E5 (3. 7 g) was subjected to silica gel CC eluting with CH_2_Cl_2_ to obtain compound **5** (4.0 mg). Compound (±)-**1** was finally separated using chiral HPLC with a Daicel chiralcel OJ-H column (5 μm, 250 × 4.6 mm, n-hexane/isopropanol = 90/10, 1 mL/min, 254 nm) to afford (−)-**1** (0.7 mg, *t_R_* = 11.7 min) and (+)-**1** (0.3 mg, *t_R_* = 11.1 min).

### 2.4. X-Ray Crystallographic Analysis

X-Ray diffraction analysis was conducted using a Bruker D8 QUEST instrument with Cu Kα radiation. The crystal structure was determined by direct methods utilizing SHELXS-97, along with difference Fourier techniques. Refinements were carried out using the program, employing full-matrix least-squares calculations on F_2_. The crystallographic data for compounds **1** and **3**, formatted according to standard CIF specifications, have been deposited with the Cambridge Crystallographic Data Centre under deposition numbers CCDC 2307473 and CCDC 2307455. Copies of these data are available free of charge upon request at the following website: www.ccdc.cam.ac.uk (accessed on 28 June 2024). Alternatively, they can be obtained by contacting the Cambridge Crystallographic Data Centre at 12 Union Road, Cambridge CB2 1EZ, UK, via fax at +44(0)1223-336-408, or by email at deposit@ccdc.cam.ac.uk.

### 2.5. ECD Calculations

The OPLS3 force field was employed for conformational analysis, and quantum chemical geometry optimization was performed at the B3LYP/6-311G (2d, q) level for each conformation. Based on the optimization results, the Boltzmann distribution of specific gravities for each conformation was obtained. Additionally, the DFT method was utilized to calculate the excited states of the dominant conformation at the CAM-B3LYP/6-311G (2d, p) level. All of the above computational jobs were performed using the Gaussian 09 package [17]. The ECD curves were calculated using SpecDis 1.70 [18] and were plotted in the Orange3 program.

### 2.6. Antimicrobial Assay

Antimicrobial activity was tested against seven microorganisms including *Staphylococcus aureus* (CMCCB26003), *Bacillus cereus* (BNCC336744), *Micrococcus luteus* (BNCC102589), *Candida albicans* (CMCC98001), *Escherichia coli* (CMCCB44102), *Pseudomonas aeruginosa* (CMCCB10104), and *Shigella Castellani* (CMCCB51572), which were obtained from Henan Engineering Research Center of Industrial Microbiology. The inhibition zone experiment was used for the initial screening of the activity of compounds **1**–**5**. We chose to use the disk diffusion assay for administration, with a dosage of 50 µg/disk. [19]. Serial dilution was carried out in 96-well plates to determine MIC values [20]. The compound concentration ranged from 0.78–100.00 μg/mL, and the results were observed 24 h after the sample was added. The minimum concentration at which no growth of the organism is shown is the MIC value. Then, the culture from the wells with significant MIC results was inoculated onto LB plates for the determination of MBC values. Due to the insolubility of the sample in water, DMSO was chosen as the solvent. The negative control was deionized water, while the positive control was ciprofloxacin.

### 2.7. Cytotoxicity Assay

The in vitro cytotoxicity was evaluated using the CCK-8 colorimetric assay against two cancer cell lines [human cervical cancer (Hela) and human thyroid cancer (KTC-1)] [21], Which were provided by Dr. Xiuming Cui. The negative control was deionized water, while the positive control was adriamycin.

## 3. Results and Discussion

### 3.1. Structure Elucidation

Compound (±)-**1** was isolated as a yellow crystal. The molecular formula of (±)-**1** was C_18_H_22_O_5_, as deduced using HR-ESI-MS (*m*/*z* 341.1357 [M + Na]^+^, calcd for 341.1359), with eight degrees of unsaturation. The IR absorption bands at 3439 and 1734 cm^−1^ suggested the presence of hydroxy and carbonyl groups. The 1D NMR spectrum (Table 1) of (±)-**1** presented the easily identified peaks of four methyls at *δ*_H_ 2.06 (s), 2.54 (s), 1.21 (d, *J* = 6.3), and 1.22 (d, *J* = 7.1); one olefinic proton at *δ*_H_ 7.33 (s); and one oxygenated proton at *δ*_H_ 4.02 (m). The ^13^C NMR and DEPT spectra data (Table 2) showed 18 carbon resonances, including four methyls (*δ*_C_ 15.4, 26.5, 16.0, and 21.8), three methylenes (*δ*_C_ 23.6, 47.7, and 48.5), three methines (*δ*_C_ 33.9, 67.4, and 129.7), and eight quaternary carbons (two carbonyl carbons at *δ*_C_ 203.1 and 205.5; five olefinic carbons at *δ*_C_ 111.0, 113.3, 117.0, 155.4, and 160.1; and one ketal carbon at *δ*_C_ 102.9). The above mentioned data showed there is a penta-substituted benzene ring (ring A) (*δ*_C_ 111.0, 113.3, 117.0, 129.7, 155.4, and 160.1) in compound **1**. The penta-substituted benzene ring was assigned using the HMBC spectrum (Figure 2), with correlations from H_3_-14 (*δ*_H_ 2.06) to C-1 (*δ*_C_ 155.4), C-2 (*δ*_C_ 117.0), C-3 (*δ*_C_ 129.7), from H-3 (*δ*_H_ 7.33) to C-1 and C-5 (*δ*_C_ 160.1), from H_3_-16 (*δ*_H_ 2.54) to C-4 (*δ*_C_ 113.3) and C-15 (*δ*_C_ 203.1), and from 5-OH (*δ*_H_ 12.83) to C-4, C-5, and C-6 (*δ*_C_ 111.0). The HMBC correlations from H_2_-7 (*δ*_H_ 2.47/2.75) to C-1, C-5, and C-6 further confirmed the substitution of acetophenone. The presence of two spin systems was confirmed via the ^1^H-^1^H COSY spectrum (Figure 2), which suggested the presence of two spin systems corresponding to H_2_-7/H-8/H_3_-17 and H_2_-12/H-13/H_3_-18. Furthermore, the HMBC correlations from H-7, H-8, Me-17, and H-10 to C-9 showed that both C-8 and C-10 were directly connected to the ketal carbon C-9. The HMBC correlations of H_2_-12 to C-11, and of H_2_-10 to C-9 and C-11, showed that C-12 and C-10 were linked through C-11. These are highly similar to previously reported data of (±)-peniphenone A, a benzannulated 6,6-spiroketal structure that was first isolated from the *Penicillium dipodomyicola* HN4-3A [2]. A detailed analysis of the NMR spectrum of (±)-**1** revealed that the only difference between (±)-peniphenone A and (±)-**1** was the absence of a methyl group on C-10 of (±)-**1**, which was further supported by the HMBC correlations of H_2_-10 (*δ*_H_ 2.78/2.82) to C-8, C-9, and C-11. In summary, the structure of compound (±)-**1** has been elucidated and named as (±)-penicrustone A (Figure 1 and Figure 2).

The single-crystal X-Ray crystallographic analysis confirmed the structure of compound 1 and determined its relative configuration (Figure 3). Through single-crystal X-Ray analysis (space group *P*21*/n*), it was found that **1** was made up of a pair of racemic compounds. The racemic nature was also supported by a subsequent HPLC analysis of (±)-**1** on a chiral phase column, where two distinct chromatographic peaks appeared in an approximate ratio of 1:2. (Appendix A). Finally, this partially racemic mixture was further separated using chiral HPLC to afford (−)-**1** (0.7 mg) and (+)-**1** (0.3 mg). Unfortunately, we only obtained the experimental CD spectrum and optical rotations of (−)-**1**{${\left[\alpha \right]}_{D}^{25}$−77.80}, but failed to obtain these for (+)-**1** due to only having a trace sample amount. ECD calculations were performed at the CAM-B3LYP/6-311G (2d, q)//6-311G (2d, p) level; the calculated ECD spectra of (8*S*, 9*R*, 13*S*)-**1** was in good agreement with the experimental data of (−)-**1** (Figure 4). The absolute configuration of (−)-**1** was assigned as 8*S*, 9*R*, 13*S*, and that of its enantiomer as 8*R*, 9*S*, 13*R*.

Compound **2** was isolated as a yellow oil. (+)-HR-ESI-MS analysis of **2** returned the molecular formula C_28_H_26_O_8_ (*m*/*z* 491.1703 [M+H]^+^, calcd for 491.1700) with sixteen degrees of unsaturation. The NMR data (Table 1) of compound **2** showed 28 carbons, including 6 methyls, 1 methylene, 3 methines, and 18 quaternary carbons. The above mentioned data showed that there were two penta-substituted benzene rings (ring A: *δ*_C_ 108.5, 115.0, 117.6, 130.0, 155.0, and 159.9; ring E: *δ*_C_ 109.9, 114.5, 117.8, 132.5, 153.8, and 158.6) in compound **2**. The penta-substituted benzene ring A was assigned according to the following HMBC correlations (Figure 2): from H_3_-21 (*δ*_H_ 1.80) to C-1, C-2, and C-3; from H-3 (*δ*_H_ 7.26) to C-1 and C-5; from H_3_-23 (*δ*_H_ 2.59) to C-4 and C-22; and from 5-OH (*δ*_H_ 12.88) to C-4, C-5, and C-6. Similarly, the penta-substituted benzene ring E was assigned according to the following HMBC correlations (Figure 2): from H_3_-26 (*δ*_H_ 1.93) to C-18, C-19, and C-20; from H-18 (*δ*_H_ 7.43) to C-16, C-20, and C-27; from H_3_-28 (*δ*_H_ 2.52) to C-17; and from 16-OH (*δ*_H_ 13.14) to C-15, C-16, and C-17. The HMBC correlations from H_2_-7 (*δ*_H_ 2.82/3.82) to C-1, C-5, C-6, C-8, C-9 (*δ*_C_ 97.9, ketal carbon), and C-24 showed that C-6 and C-7 were directly connected, as well as demonstrating the presence of ring B (formed by C-1/C-6/C-7/C-8/C-9/O). On the basis of the HMBC correlations of H_3_-24 to C-7, C-8, C-9, and C-12, these showed that C-7 and C-12 were connected through C-8. Correlations from H-11 (*δ*_H_ 7.45) to C-9, C-8, and C-13, as well as those from H_3_-25 (*δ*_H_ 2.24) to C-10, C-13, and C-14, connect the last two units (ring C and D). Due to the chemical shifts of C-1, C-14, and C-9, as well as the unsaturation requirement of **2**, C-1 and C-14 should be connected to C-9 through an oxygen atom, respectively, and C-10 should be connected to C-9. Considering the unsaturation requirements of the structure, C-14 was linked to C-15. On the basis of these data, Compound **2** was established as shown, and was named as penicrustone B. The calculated ECD for 8*S*, 9*S* of **2** match the experimental curve used to determine the absolute configuration (Figure 3).

Compound (±)-**3** was isolated as a colorless crystal. According to the HR-ESI-MS (*m*/*z* 261.1244 [M-H]-, calcd for 261.1245), the molecular formula of (±)-**3** was C_14_H_18_N_2_O_3_, and the degree of unsaturation was seven. The NMR data (Table 1) gave 14 carbon resonances that were classified into 2 methyls, 4 methylenes, 2 methines, and 6 quaternary carbons. The above mentioned data showed that there is a penta-substituted benzene ring (*δ*_C_ 110.6, 112.7, 117.9, 132.1 161.8, and 163.6) in compound (±)-**3**. The penta-substituted benzene ring (ring A) was assigned according to the HMBC spectrum (Figure 2), with the correlations from H_3_-12 (*δ*_H_ 2.18) to C-1, C-2, and C-3; from H-3 (*δ*_H_ 7.43) to C-1, C-2, and C-5; and from H_3_-14 (*δ*_H_ 2.54) to C-4 and C-13 (*δ*_C_ 202.7). The HMBC correlations from H_2_-7 (*δ*_H_ 4.30) to C-1 and C-5 further confirmed the substitution of acetophenone. The ^1^H-^1^H COSY spectrum (Figure 2) suggested the presence of one spin system corresponding to H-8/H_2_-9/H_2_-10/H_2_-11. Considering the chemical shifts of C-7 (*δ*_C_ 50.7), C-8 (*δ*_C_ 143.5), and C-11 (*δ*_C_ 47.2), the unsaturation requirement, combined with the HR-ESI-MS spectrometry of compound **3** for analyses, revealed the presence of ring B. The HMBC correlations of H_2_-7 to C-1 and C-11 showed that ring A and ring B were linked through C-7. The X-Ray crystal structure of **3** is shown in Figure 3. However, the crystal structure of **3** occurs as a racemate, as evidenced by the centrosymmetric space group *P*21*/c*. The racemic nature was also supported by a lack of optical rotation and CD (circular dichroism) maximum (Appendix A). Finally, this racemic mixture was further separated according to chiral HPLC; unfortunately, it could not be separated. Due to the presence of nitrogen atoms, compound **3** can undergo an inversion of its lone pair electrons, leading to a change in configuration. This results in chirality, despite the absence of a chiral carbon, which is consistent with the characteristics described in the literature [22,23]. Finally, the new compound (±)-**3** was named as (±)-penicrustone C.

Compound **4** was a yellow oil. The molecular formula of 4 is C_16_H_20_O_4_, as deduced by the HR-ESI-MS (*m*/*z* 275.1288 [M − H]^−^, calcd for 275.1289), with seven degrees of unsaturation. A detailed comparison of 1D and 2D NMR spectroscopic data of **4** (Table 2) with compounds **1**–**3** suggested that compound **4** also has 1,5-dihydroxy-2-methylacetophenone moieties (*δ*_C_ 160.6, 118.0, 130.6, 112.9, 161.9, 113.7, 16.3, 202.9, and 26.4). The remaining seven carbon signals were assigned according to the ^1^H-^1^H COSY correlations of H_2_-7/H-8/H_3_-16 and H-10/H-11/H_3_-12, and the HMBC correlations from H_3_-16/ H-11/H-7 to C-9. The allocation of planar structures is achieved through the connection of C-6 and C-7, which is supported by the key HMBC correlations from H-7 to C-1, C-5, and C-6. The double bond was assigned as having a trans geometry based on the big coupling constant (15.5) of H-10 and H-11.

Subsequently, by comparing the calculated and experimental ECD data (Figure 3), the absolute configuration of 4 is designated as 8*S*. The above information indicated that **4** was similar to communol E [24]. Therefore, compound **4** was constructed as described, and was given a name—penicrustone D.

Compound **5** was isolated as a yellow oil and possessed a molecular formula of C_16_H_22_O_4_, as assigned according to the HRESIMS, indicating six degrees of unsaturation. NMR data (Table 2) showed 16 carbon resonances, including 4 methyls, 3 methylenes, 2 methines, and 7 quaternary carbons. These data showed a significant similarity to those of compound **4**. These data indicated that compound **5** possessed a similar structure to that of 4, except that the double bond between C-10 and C-11 was not present in **5** (Figure 1 and Figure 2). In compound 5, C-10 (*δ*_C_ 43.0) and C-11 (*δ*_C_ 17.4)—two methylene groups—are present, which are not present in its counterpart in compound 4, which was supported by the DEPT and HSQC. The absolute configuration of **5** was deduced as 8*S* upon comparison with the calculated ECD spectrum of **5** and the experimental one (Figure 3). Therefore, compound **5** was elucidated as shown in Figure 1, and was named as penicrustone E.

### 3.2. Compound Characterization

(±)-Penicrustone A [(±)-**1**]: yellow crystals (MeOH); mp 107 °C; ${\left[\alpha \right]}_{D}^{25}$−82.74 (c 0.48, MeOH); UV (MeOH) λ_max_ (log ε) 217 (3.4), 280 (3.3), 328 (2.9) nm; IR (KBr) ν_max_ 3439, 3224, 2976, 2934, 1734, 1627, 1476, 1381 cm^−1^; ^1^H NMR (600 MHz, CDCl_3_) and ^13^C NMR (150 MHz, CDCl_3_) data, see Table 1; HREIMS *m*/*z* 341.1357 [M + Na]^+^ (calcd for C_18_H_22_O_5_Na^+^: 341.1359).

(−)-Penicrustone A [(−)-**1**]: ${\left[\alpha \right]}_{D}^{25}$−77.80 (c 0.10, MeOH)

Penicrustone B (**2**): yellow oil; ${\left[\alpha \right]}_{D}^{25}$−7.20 (c 0.10, MeOH); UV (MeOH) λ_max_ (log ε) 215 (3.4), 282 (3.3), 336 (3.0), 372 (3.0) nm; IR (KBr) ν_max_ 3415, 3073, 2875, 2860, 1632, 1377, 1313, 1182 cm^−1^; ^1^H NMR (600 MHz, CDCl_3_) and ^13^C NMR (150 MHz, CDCl_3_) data, see Table 1; HRESIMS *m*/*z* 491.1703 [M+H]^+^(calcd for C_28_H_27_O_8_: 491.1700).

(±)-Penicrustone C (**3**): colorless crystals (MeOH); mp 108 °C; ${\left[\alpha \right]}_{D}^{25}$ ±0 (c 0.10, MeOH); UV (MeOH) λ_max_ (log ε) 203 (2.8), 211 (2.7), 283 (2.1), 329 (1.8) nm; IR (KBr) ν_max_ 3434, 2924, 1732, 1626, 1186, 968 cm^−1^; ^1^H NMR (600 MHz, CDCl_3_) and ^13^C NMR (150 MHz, CDCl_3_) data, see Table 1; HRESIMS *m*/*z* 261.1244 [M − H]^−^ (calcd for C_14_H_17_N_2_O_3_: 261.1245).

Penicrustone D (**4**): pale yellow oil; ${\left[\alpha \right]}_{D}^{25}$+1.00 (c 0.12, MeOH); UV (MeOH) λ_max_ (log ε) 218 (3.5), 283 (3.1), 330 (2.8) nm; IR (KBr) ν_max_ 3415, 3251, 1730, 1679, 1624, 1445, 1375 cm^−1^; ^1^H NMR (600 MHz, CDCl_3_) and ^13^C NMR (150 MHz, CDCl_3_) data, see Table 2; HRESIMS m/z 275.1288 [M − H]^−^ (calcd for C_16_H_19_O_4_: 275.1289).

Penicrustone E (**5**): pale yellow oil; ${\left[\alpha \right]}_{D}^{25}$−6.30 (c 0.40, MeOH); UV (MeOH) λ_max_ (log ε) 217 (3.7), 231 (3.4), 283 (3.6), 328 (3.2) nm; IR (KBr) ν_max_ 3443, 3242, 2964, 2934, 1720, 1692, 1627, 1382 cm^−1^; ^1^H NMR (600 MHz, CDCl_3_) and ^13^C NMR (150 MHz, CDCl_3_) data, see Table 2; HRESIMS *m*/*z* 277.1443 [M − H]^−^ (calcd for C_16_H_21_O_4_: 277.1445).

### 3.3. X-Ray Crystallographic Analysis of Compound **1** and **3**

Crystal data for (±)-**1**: C_18_H_22_O_5_, yellow crystals, was obtained from a solution of EtOAc. M = 318.35, monoclinic, a = 7.4454 (2) Å, b = 15.5624 (4) Å, c = 14.1811 (3) Å, α = 90°, β = 103.1380 (10)^°^, γ = 90°, V = 1600.13 (7) Å^3^, T = 150.(2) K, space group P21/n, Z = 4, μ (Cu Kα) = 0.788 mm^−1^, 16161 reflections measured, 3140 independent reflections (R_int_ = 0.0575). The final R_1_ value was 0.0396 [>2σ(I)]. The final wR (F^2^) value was 0.0964 [I > 2σ(I)]. The final R_1_ value was 0.0462 (all data). The final wR (F^2^) value was 0.1006 (all data). The goodness of fit on F^2^ was 1.063.

Crystallographic data for (±)-**3**: C_14_H_18_N_2_O_3_, colorless crystals obtained from a solution of MeOH. M = 262.30, monoclinic, a = 14.5626 (6) Å, b = 10.4114 (5) Å, c = 8.7630 (4) Å, α = 90°, β = 94.952 (2)°, γ = 90°, V = 1323.66 (10) Å^3^, T = 150.(2) K, space group P21/c, Z = 4, μ (Cu Kα) = 0.764 mm^−1^, 19943 reflections measured, 2585 independent reflections (R_int_ = 0.0529). The final R_1_ value was 0.0382 [I > 2σ(I)]. The final wR (F^2^) value was 0.1007 [I > 2σ(I)]. The final R_1_ value was 0.0402 (all data). The final wR (F^2^) value was 0.1024 (all data). The goodness of fit on F^2^ was 1.045.

### 3.4. Biological Activities

The activity of isolated compounds **1**–**5** against seven microorganisms was studied. The preliminary screening of antimicrobial activity against strains used compounds at 50 µg/disk. The results showed that compounds **4** and **5** had good inhibitory effects (Table 3).

Subsequently, the MIC and MBC values of 4 and 5 were tested; the results are shown in Table 4. Compounds **4** and **5** have good inhibitory effects on *Micrococcus luteus*, with MIC values of 12.5 and 25 μg/mL, respectively. Comparing the MIC and MBC values of compounds **4** and **5**, it was found that the inhibitory effect of the compounds on Gram-positive bacteria were superior to that on Gram-negative bacteria.

The cytotoxicity of compounds **1**–**5** on tumor cell lines Hela and KTC-1 was tested using the CCK-8 method. As shown in Table 5, compounds **4** and **5** exhibit significant cytotoxicity, with IC_50_ values ranging from 4.26 to 28.84 μg/mL. Compound **4** has a strong inhibitory effect on Hela and KTC-1 cell lines, with IC_50_ values of 4.26 and 4.28 μg/mL. Compound **5** has weak cytotoxicity on Hela and KTC-1, with IC_50_ values of 28.84 and 23.85 μg/mL, respectively. The other three compounds did not show significant cytotoxicity.

## 4. Conclusions

To derive more compounds with novel structures, we investigated the metabolites of *P. crustosum* and found five new unreported compounds; their structures were elucidated on the basis of extensive spectroscopic analysis, including 1D and 2D NMR, as well as HR-ESI-MS. Their absolute configurations were determined via ECD calculations and single-crystal X-Ray crystallography, which enriched the chemical constitution of *P. crustosum.* Biological activity assays showed that compounds **4** and **5** exhibited good inhibitory activity against *Micrococcus luteus*, as well as cytotoxicity against tumor cell lines Hela and KTC-1.

## Figures and Tables

**Figure 1 microorganisms-12-02195-f001:**
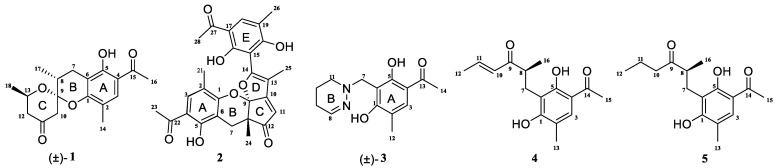
Structures of compounds **1**–**5**; (±) represents racemic form.

**Figure 2 microorganisms-12-02195-f002:**
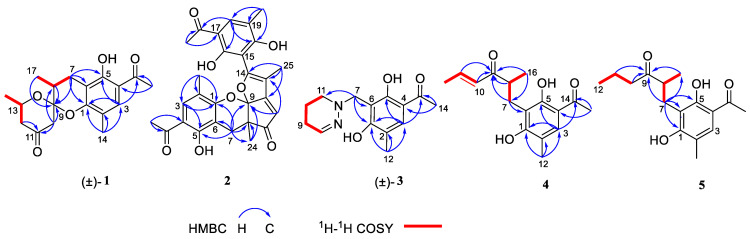
Key HMBC and ^1^H-^1^H COSY correlations of compounds **1**–**5**; (±) represents racemic form.

**Figure 3 microorganisms-12-02195-f003:**
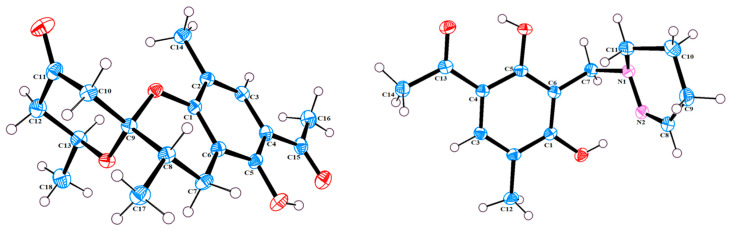
X-Ray crystallographic structure of **1** and **3**.

**Figure 4 microorganisms-12-02195-f004:**
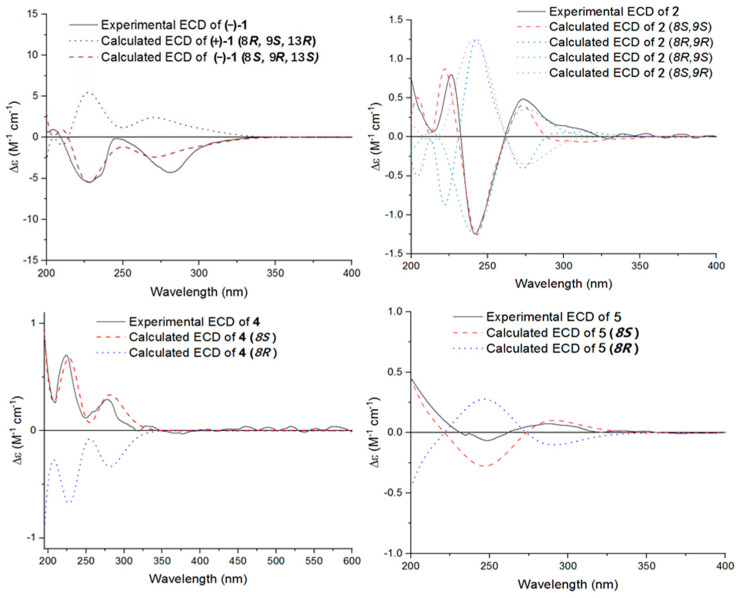
Experimental and calculated ECD curves of compounds (−)-**1**, **2**, **4**, and **5**.

**Table 1 microorganisms-12-02195-t001:** ^1^H NMR and ^13^C NMR spectroscopic data of compounds **1**–**3** in CDCl_3_ (600 MHz, TMS).

No.	1		2		3	
	*δ*_C_, Type	*δ*_H_, Mult. (*J* in Hz)	*δ*_C_, Type	*δ*_H_, Mult. (*J* in Hz)	*δ*_C_, Type	*δ*_H_, Mult. (*J* in Hz)
1	155.4, C		155.0, C		161.8, C	
2	117.0, C		117.6, C		117.9, C	
3	129.7, CH	7.33, s	130.0, CH	7.26, s	132.1, CH	7.43, s
4	113.3, C		115.0, C		112.7, C	
5	160.1, C		159.9, C		163.6, C	
6	111.0, C		108.5, C		110.6, C	
7	23.6, CH_2_	2.50, dd (16.1, 3.1)	22.4, CH_2_	2.82, d, (17.0)	50.7, CH_2_	4.30, s
		2.76, dd (16.1, 5.6)		3.82, d, (17.0)		
8	33.9, CH	1.95, m	47.5, C		143.5, CH	6.88, brs
9	102.9, C		97.9, C		22.5, CH_2_	2.12, m
10	47.7, CH_2_	2.56, d (14.2)	126.3, C		20.0, CH_2_	2.91, m
		2.78, d (14.2)				
11	205.5, C		117.6, CH	7.45, s	47.2, CH_2_	1.98, m
12	48.5, CH_2_	2.30, m	193.1, C		16.0, CH_3_	2.18, s
		2.48, m				
13	67.4, CH	4.02, m	123.0, C		202.7, C	
14	15.4, CH_3_	2.06, s	142.5, C		26.4, CH_3_	2.54, s
15	203.1, C		109.9, C			
16	26.5, CH_3_	2.54, s	158.6, C			
17	16.0, CH_3_	1.22, d (7.1)	114.5, C			
18	21.8, CH_3_	1.21, d (6.3)	132.5, CH	7.43, s		
19			117.8, C			
20			153.8, C			
21			15.5, CH_3_	1.80, s		
22			203.2, C			
23			26.7, CH_3_	2.59, s		
24			24.9, CH_3_	1.47, s		
25			11.2, CH_3_	2.24, s		
26			14.6, CH_3_	1.93, s		
27			203.5, C			
28			26.5, CH_3_	2.52, s		
OH-5		12.83, s		12.88, s		13.04, s
OH-16				13.14, s		

**Table 2 microorganisms-12-02195-t002:** ^1^H NMR and ^13^C NMR spectroscopic data of compounds **4**–**5** in CDCl_3_ (600 MHz, TMS).

No.	4		5	
	*δ*_C_, Type	*δ*_H_, Mult. (*J* in Hz)	*δ*_C_, Type	*δ*_H_, Mult. (*J* in Hz)
1	160.6, C		160.4, C	
2	118.0, C		117.9, C	
3	130.6, CH	7.34, s	130.6, CH	7.35, s
4	112.9, C		113.5, C	
5	161.9, C		161.8, C	
6	113.7, C		112.9, C	
7	25.3, CH_2_	2.73, dd (14.3, 2,6)	25.3, CH_2_	2.70, dd (14.5, 2.7)
		2.83, dd (14.3, 10.7)		2.78, dd (14.5, 10.6)
8	44.6, CH	3.27, m	46.9, CH	3.08, m
9	207.3, C		220.1, C	
10	130.0, CH	6.18, d (15.5)	43.0, CH_2_	2.32, dt (16.8, 7.0)
				2.58, overlapped
11	146.0, CH	7.00, m	17.4, CH_2_	1.53, q (7.3)
12	18.6, CH_3_	1.89, d (7.1)	13.7, CH_3_	0.81, t (7.4)
13	16.3, CH_3_	2.18, s	16.2, CH_3_	2.18, s
14	202.9, C		202.9, C	
15	26.4, CH_3_	2.52, s	26.4, CH_3_	2.52, s
16	19.4, CH_3_	1.31, d (7.4)	18.6, CH_3_	1.31, d (7.4)
OH-5		9.18, s		8.90, s

**Table 3 microorganisms-12-02195-t003:** Antimicrobial activity of compounds **1**–**5** (zone of inhibition, mm).

Microorganisms	Compounds (50 µg/Disk)							
	1	2	3	4	5	DW	DMSO	Ciprofloxacin
*Pseudomonas aeruginosa*	NA	NA	NA	21.20 ± 0.06	6.60 ± 0.03	NA	NA	34.17 ± 0.09
*Escherichia coli*	NA	NA	NA	11.13 ± 0.26	6.97 ± 0.08	NA	NA	35.30 ± 0.03
*Shigella Castellani*	NA	NA	NA	26.82 ± 0.43	6.63 ± 0.03	NA	NA	34.41 ± 0.54
*Staphylococcus aureus*	NA	NA	NA	12.00 ± 0.05	7.80 ± 0.03	NA	NA	31.50 ± 0.06
*Bacillus cereus*	NA	NA	NA	12.40 ± 0.15	9.80 ± 0.27	NA	NA	27.20 ± 0.10
*Micrococcus luteus*	NA	NA	NA	11.00 ± 0.08	8.80 ± 0.05	NA	NA	34.87 ± 0.07
*Candida albicans*	NA	NA	NA	11.00 ± 0.05	9.03 ± 0.38	NA	NA	35.27 ± 0.27

NA: no activity; DW: deionized water; DMSO is used to eliminate solvent interference in experiments; values represent the mean of three replicates ± SD.

**Table 4 microorganisms-12-02195-t004:** Antimicrobial activity of compounds 1–5 (MIC and MBC, μg/mL).

Microorganisms	Compounds					
	4	5	Ciprofloxacin	4	5	Ciprofloxacin
	MIC			MBC		
*Pseudomonas aeruginosa*	50 ± 0	>100	3.12 ± 1.81	100 ± 0	>100	6.25 ± 0
*Escherichia coli*	25 ± 0	>100	6.25 ± 0	50 ± 0	>100	12.5 ± 0
*Shigella Castellani*	25 ± 0	>100	≤0.78	50 ± 0	>100	1.56 ± 0
*Staphylococcus aureus*	50 ± 0	100 ± 0	1.56 ± 0	100 ± 0	>100	3.12 ± 1.81
*Bacillus cereus*	25 ± 0	100 ± 0	≤0.78	100 ± 0	>100	≤0.78
*Micrococcus luteus*	12.50 ± 0	25 ± 0	≤0.78	50 ± 0	50 ± 0	1.56 ± 0
*Candida albicans*	50 ± 0	>100	≤0.78	>100	>100	≤0.78

Values represent the mean of three replicates ± SD.

**Table 5 microorganisms-12-02195-t005:** Cytotoxic activity of compounds **1**–**5** (IC_50_ in μg/mL).

Cell Line	Compounds					Adriamycin
	1	2	3	4	5	(IC_50_ in μg/mL)
Hela	>50	>50	>50	4.64 ± 0.32	28.84 ± 0.16	0.03 ± 0.08
KTC-1	>50	>50	>50	4.28 ± 0.53	23.85 ± 0.85	0.04 ± 0.13

Values represent the mean of three replicates ± SD.

## Data Availability

The original contributions presented in the study are included in the article; further inquiries can be directed to the corresponding author due to privacy.

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
