# Peer review of "Five Unreported Ketone Compounds—Penicrustones A–E—From the Endophytic Fungus Penicillium crustosum"

_microorganisms, 2024, doi:10.3390/microorganisms12112195_

Round 1
Reviewer 1 Report
Comments and Suggestions for Authors
This study investigates the isolation of five ketones, penicrustones A-E, from the solid fermentation of Penicillium crustosum. however some issues must be improved prior publication.
There are not statistical analysis.
The antimicrobial assay must be improved. I suggest including MBC and even measure the diameter of inhibition zone vs positive and negative control.
The chemical structures of these compounds were elucidated via X-ray crystallographic analysis, NMR spectroscopy, ESI-MS and HR-ESI-MS analysis, and UV spectroscopy. Then the antibacterial properties of pericrustones were estimated.
Author Response
Comments 1: There are not statistical analysis.
Response 1:
Thank you for pointing this out. We agree with this comment. Therefore, we have redone the antimicrobial experiment, and the activity experiment data has been subjected to statistical analysis again. Please refer to Tables 3, 4, and 5 on pages 9 to 10 for details.
Comments 2: The antimicrobial assay must be improved. I suggest including MBC and even measure the diameter of inhibition zone vs positive and negative control.
Response 2:
Based on your suggestion, we have improved and re conducted the antibacterial experiment, including the inhibition zone experiment, MIC and MBC determination, with three replicates each, and the results have been statistically analyzed. And it was described in the manuscript. In the revised manuscript this change can be found – page 3, section 2.6 and page 9, section 3.4. Experimental photos can be seen in the Supplementary Materials: Figure S47.
Comments 3: The chemical structures of these compounds were elucidated via X-ray crystallographic analysis, NMR spectroscopy, ESI-MS and HR-ESI-MS analysis, and UV spectroscopy. Then the antibacterial properties of pericrustones were estimated.
Response 3:
The structure of the compounds has been elaborated in detail in the article (pages 3 to 8, section 3.1), followed by testing of their antibacterial and cytotoxic (pages 3, sections 2.6 and 2.7), and finally a summary has been made (page 10, section 4).
Reviewer 2 Report
Comments and Suggestions for Authors
The study describes structure and biological activities of 5 penicustone compounds isolated from a Penicillium crustosum strain. The chemical structure of the purified compounds are elucidated and the moderate activity against selected bacterial strains and cancer cell lines are described.
All new bioactive fungal metabolites are of interest, and this article provides novel information of such metabolites (according to my understanding). I think it was interesting to test the compounds on bacteria, fungi and mammalian cells, and compare activity.
I am not qualified to comment on the chemical structures reported, but to me they seems adequately performed.However, I have some comments on the biological activity testing.
1. For the bacteriological and cell testing: What are the nonspecific upper limit of the assay, for the bacteriological assay the range of tested concentrations is given as 100 ug/ml, but for the cell assays no such range is given. Where the positive control given seems adequate, the negative control is DMSO and not an purified inactive fungal metabolite?. May be the choiceof of negative control could be commented and explained by the authors.
2. In Table 4 the concentrations of the tested metabolites are expressed in uM/ml but in Table 3 in ug/ml. It would be easier for the reader if both units are given in both tables.
3. The cytotoxicity is tested only on malign cell lines. The authors could explain and discuss shortly the metabolic difference and possible difference in sensitivity between malign and primary mammalian cell lines, or at least think about it.
4. The choice of microbial test strains?, the authors claim they are pathogenic, but do not provide any information of the pathogenicity of the particular bacterial and fungal strains tested, were they clinical isolates or just representing potentially pathogenic species?. The authors should provide that information.
Author Response
Comments 1: For the bacteriological and cell testing: What are the nonspecific upper limit of the assay, for the bacteriological assay the range of tested concentrations is given as 100 μg/ml, but for the cell assays no such range is given. Where the positive control given seems adequate, the negative control is DMSO and not a purified inactive fungal metabolite? May be the choice of negative control could be commented and explained by the authors.
Response 1: Thank you for pointing this out. We have made corrections to the relevant content based on your suggestions.
(1) We have reanalyzed the cytotoxicity results and provided a non-specific upper limit of 50 ug/ml. Please refer to Tables 5 on page 10 for details.
(2) First, since the samples were insoluble in water, DMSO was chosen as the solvent. During the experiment, deionized water and DMSO were set up as negative control groups, with the DMSO group used to exclude solvent interference in the experiment. The negative control has now been corrected to deionized water, which has no effect on the activity experiments. Please refer to sections 2.6 and 2.7 on page 3 for details.
Comments 2: In Table 4 the concentrations of the tested metabolites are expressed in μM/ml but in Table 3 in μg/ml. It would be easier for the reader if both units are given in both tables.
Response 2:
Thank you for pointing this out. Based on your suggestion, we have standardized the units in the table to ug/ml and conducted statistical analysis. Please refer to Tables 4 and 5 on pages 9 to 10 for details.
Comments 3: The cytotoxicity is tested only on malign cell lines. The authors could explain and discuss shortly the metabolic difference and possible difference in sensitivity between malign and primary mammalian cell lines, or at least think about it.
Response 3:
We greatly appreciate your suggestions. We believe that your recommendations are highly constructive for our experiments, and we will certainly address this issue in our future work. Due to our oversight in the initial stages, we did not prepare primary cells. However, the results of our completed cytotoxicity experiments have already demonstrated that Compound 4 and 5 exhibit inhibitory effects on both tumor cell lines. We designed our cell experiments by referencing the following literature [1-3], and we hope this can serve as a response to your valuable suggestion.
Comments 4: The choice of microbial test strains? the authors claim they are pathogenic, but do not provide any information of the pathogenicity of the particular bacterial and fungal strains tested, were they clinical isolates or just representing potentially pathogenic species? The authors should provide that information.
Response 4:
The microbial test strains are all potential pathogenic strains purchased from the Henan Engineering Research Center of Industrial Microbiology, not clinical isolates. The detailed information of the strains has been provided in the manuscript, and the human pathogenic strains in the text have been corrected to microorganisms. For details, please refer to section 2.6 on page 3.
- Liu, C.C.; Zhang, Z.Z.; Feng, Y.Y.; Gu, Q.Q.; Li, D.H.; Zhu, T.J. Secondary metabolites from Antarctic marine-derived fungus Penicillium crustosum HDN153086. Nat Prod Res 2019, 33, 414-419, doi:10.1080/14786419.2018.1455045.
- Du, K.; Zhang, Z.; Jing, D.; Wang, Y.; Li, X.; Meng, D. Diterpene glycosides, acetophenone glycosides and tannins from polar extracts of the root of Euphorbia fischeriana with cytotoxicity and antibacterial activities. Phytochemistry 2022, 203, doi:10.1016/j.phytochem.2022.113382.
- Xu, L.-L.; Chen, H.-L.; Hai, P.; Gao, Y.; Xie, C.-D.; Yang, X.-L.; Abe, I. (+)- and (−)-Preuisolactone A: A Pair of Caged Norsesquiterpenoidal Enantiomers with a Tricyclo[4.4.01,6.02,8]decane Carbon Skeleton from the Endophytic Fungus Preussia isomera. Organic Letters 2019, 21, 1078-1081, doi:10.1021/acs.orglett.8b04123.
Round 2
Reviewer 1 Report
Comments and Suggestions for Authors
Agree with the revised manuscript.
Reviewer 2 Report
Comments and Suggestions for Authors
I have no further comments or suggestions